# The potential of gene drives in malaria vector species to control malaria in African environments

Penelope A. Hancock ®[1,7] ✉, Ace North ®[2,7], Adrian W. Leach ®[3], Peter Winskill ®[1], Azra C. Ghani ®[1], H. Charles J. Godfray ®[2,4], Austin Burt ®[5] & John D. Mumford ®[3,6]

Gene drives are a promising means of malaria control with the potential to cause sustained reductions in transmission. In real environments, however, their impacts will depend on local ecological and epidemiological factors. We develop a data-driven model to investigate the impacts of gene drives that causes vector population suppression. We simulate gene drive releases in sixteen ~ 12,000 km$^2$ areas of west Africa that span variation in vector ecology and malaria prevalence, and estimate reductions in vector abundance, malaria prevalence and clinical cases. Average reductions in vector abundance ranged from 71.6–98.4% across areas, while impacts on malaria depended strongly on which vector species were targeted. When other new interventions including RTS,S vaccination and pyrethroid-PBO bednets were in place, at least 60% more clinical cases were averted when gene drives were added, demonstrating the benefits of integrated interventions. Our results show that different strategies for gene drive implementation may be required across different African settings.

The global burden of malaria has substantially reduced since the turn of the century, yet in recent years progress has stalled and the annual rate of malaria-related deaths currently exceeds that in 2019[1]. There is an urgent need for new approaches, and one of the most promising options is to use gene drive technologies[2]. Gene drives are genetic elements that bias their own inheritance above what is predicted by Mendelien genetics, enabling their spread in populations where introduced[3–5]. They may be used in malaria control either to suppress vector populations by inhibiting an essential gene or to modify populations by expressing a gene that reduces disease transmission. The past decade has seen rapid progress in gene drive research, spurred by major advances in CRISPR-cas9 technology, bringing us to a point where both suppression drives[6,7] and modification drives[8,9] have been created in laboratory populations of *Anopheles* mosquitoes[10].

Field trials to test these technologies in natural settings are now being actively considered[11,12].

Mathematical models have helped to develop a theoretical understanding of how gene drive releases could impact vector populations and reduce disease prevalence. The potential for gene drives to suppress vector populations was initially investigated using analytical models of diploid, sexually reproducing populations with no spatial structure[2,13]. The models showed that suppression will be greatest when the drive element has no effect on fitness in the heterozygote (one-copy) state, and when the target is a female trait such as egg production. These insights have been instrumental in the recent development of CRISPR-cas9 gene drives targeting female fertility[7,14]. The gene drive described by Kryou et al.[7], for example, has been engineered to target a female-specific exon in the doublesex (*dsx*) gene

[1]MRC Centre for Global Infectious Disease Analysis, School of Public Health, Imperial College London, London, UK. [2]Department of Biology, University of Oxford, Oxford, UK. [3]Centre for Environmental Policy, Imperial College London, Ascot, UK. [4]Oxford Martin School, University of Oxford, Oxford, UK. [5]Department of Life Sciences, Imperial College London, Ascot, UK. [6]Deceased: John D. Mumford. [7]These authors contributed equally: Penelope A. Hancock, Ace North. ✉e-mail: p.hancock@imperial.ac.uk

in *Anopheles gambiae* mosquitoes, a highly conserved gene that is essential for sex determination and fertility. The gene drive construct has since been shown to spread rapidly through large cage populations of *An. gambiae* resulting in full population suppression[6].

Models have also explored how spatial structure will affect the impact of gene drives on vector populations[15–20]. A key insight is that a suppression gene drive may not lead to extinction of the target species across an entire landscape, even if it tends to eliminate populations locally. This is because the target species may become extinct in some parts of a landscape while persisting in other parts, where the gene drive may be absent. Over time the extant populations can recolonise the habitat where local elimination has occurred, yet if the gene drive remains in some other parts of the landscape it may eventually return and re-suppress. These spatial dynamics will be influenced by numerous factors including seasonality[16–18], dispersal propensity[19], the frequency of inbreeding and strength of inbreeding depression[19,20], and the specifics of the gene drive construct[19,21]. Gene drive spread is predicted to be most disrupted in environments where the mosquito population is composed of sparsely distributed subpopulations[15,16,18], where there is a high degree of seasonality[16,18], or where the populations have a high tendency to inbreed[19,20].

In anticipation of the first gene drive field releases, models have increasingly been customised to specific malaria-endemic locations[16–18,22–24]. Spatially explicit models that include human malaria infections have considered the impacts of both population suppression[16,22] and replacement[23,25] gene drive strategies on malaria control, alone and in combination with insecticide-treated bednet (ITN) usage. Simulations of population suppression using a driving-Y construct (which spreads because males produce more Y than X gametes) in a series of 625 km² areas within the Democratic Republic of Congo predicted that the most effective gene drives could eliminate malaria in areas where the target vector species was responsible for all malaria transmission[22]. Versions of the same model investigated releases of gene drives for vector population modification in a 300 km² area in Burkina Faso, and predicted that malaria elimination would be

possible when gene drives were combined with high rates of ITN usage, assuming the gene drive caused a 70% reduction in mosquito-to-human transmission[23].

To further our understanding of how gene drives could perform in a variety of African settings, we develop a new approach to modelling gene drive impacts that incorporates location-specific details of vector ecology and malaria epidemiology. We extend the large-scale modelling approach of North et al.[18] to incorporate high resolution geographic data layers representing human settlements, rivers and lakes, vector population abundance, and species composition. We combine this with a model of malaria transmission dynamics that estimates transmission intensity in a region accounting for changes in the coverage of vector control and human treatment interventions over time[26,27]. These include mainstay interventions such as Indoor Residual Spraying (IRS), standard pyrethroid-only insecticide-treated bednets (ITNs) and drugs for malaria treatment and prevention in humans as well as newer approaches including new types of ITNs[28] and vaccines[29].

We predict the impacts of releasing population suppression gene drives targeting the *dsx* gene on vector populations, malaria prevalence and clinical cases in sixteen areas, each ~12,000 km², located across West Africa. Female mosquitoes that are homozygous for the gene drive are sterile and unable to blood feed or transmit the malaria parasite. In each area we model gene drive releases in each of three vector species groups in the genus *Anopheles*: *An. funestus*, *An. arabiensis* and the closely related species pair *An. gambiae* and *An. coluzzii*. We investigate the factors affecting the extent to which gene drive releases suppress vector populations, including the fitness cost incurred by the gene drive as well as variables affecting population spatial structure in different landscapes. We then estimate the impacts on malaria burden depending on which species group is targeted. Motivated by concerns about transmission by other vector species that are less well surveyed or invading, we investigate how impacts on malaria depend on the presence of an additional vector species that is not one of the target species considered in our analyses. Other species from the *An. gambiae* complex contribute substantially to

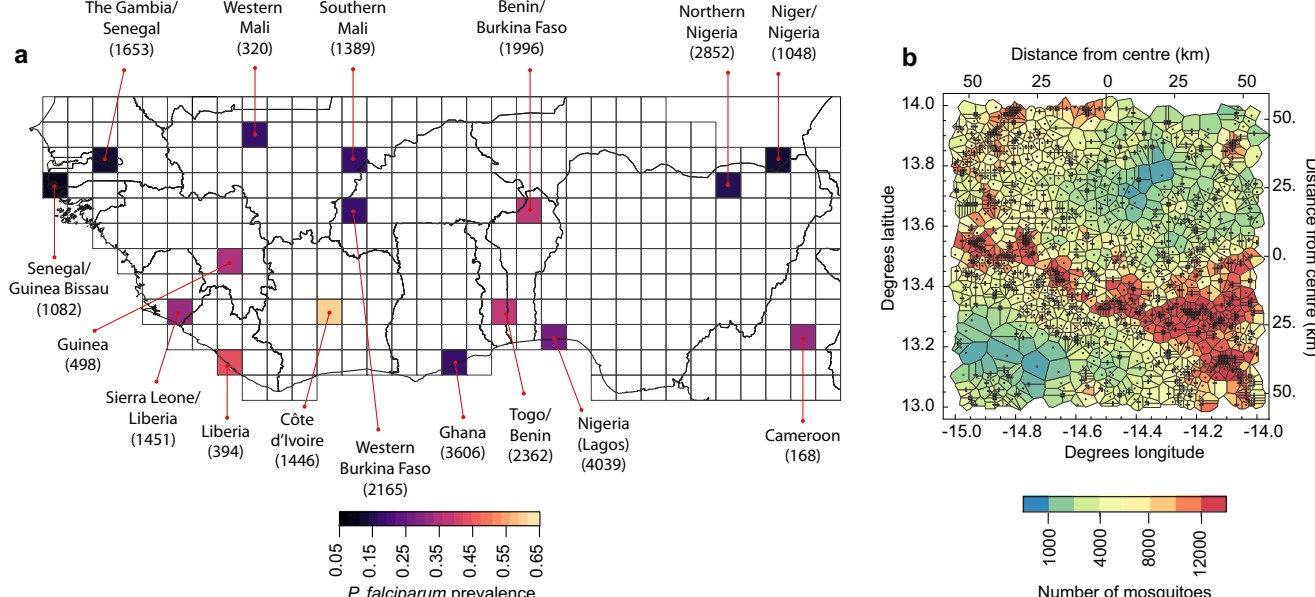

**Fig. 1 | The areas of west Africa in which we model the impacts of gene drive releases. a** The sixteen areas are coloured according to the estimated prevalence of *P. falciparum* malaria in humans residing in the area in 2019[35] (see Methods). Each area is labelled according to the countries within the area, and numbers in brackets show the estimated number of settlements in the area. Grid lines divide the region

into areas of 1° in latitude by 1° in longitude. **b** An example of the landscape model for The Gambia/Senegal area showing the estimated settlement locations (black dots), localities (polygons) and the estimated number of mosquitoes associated with each settlement. Data show the estimated annual maximum number of mosquitoes in the year 2018.

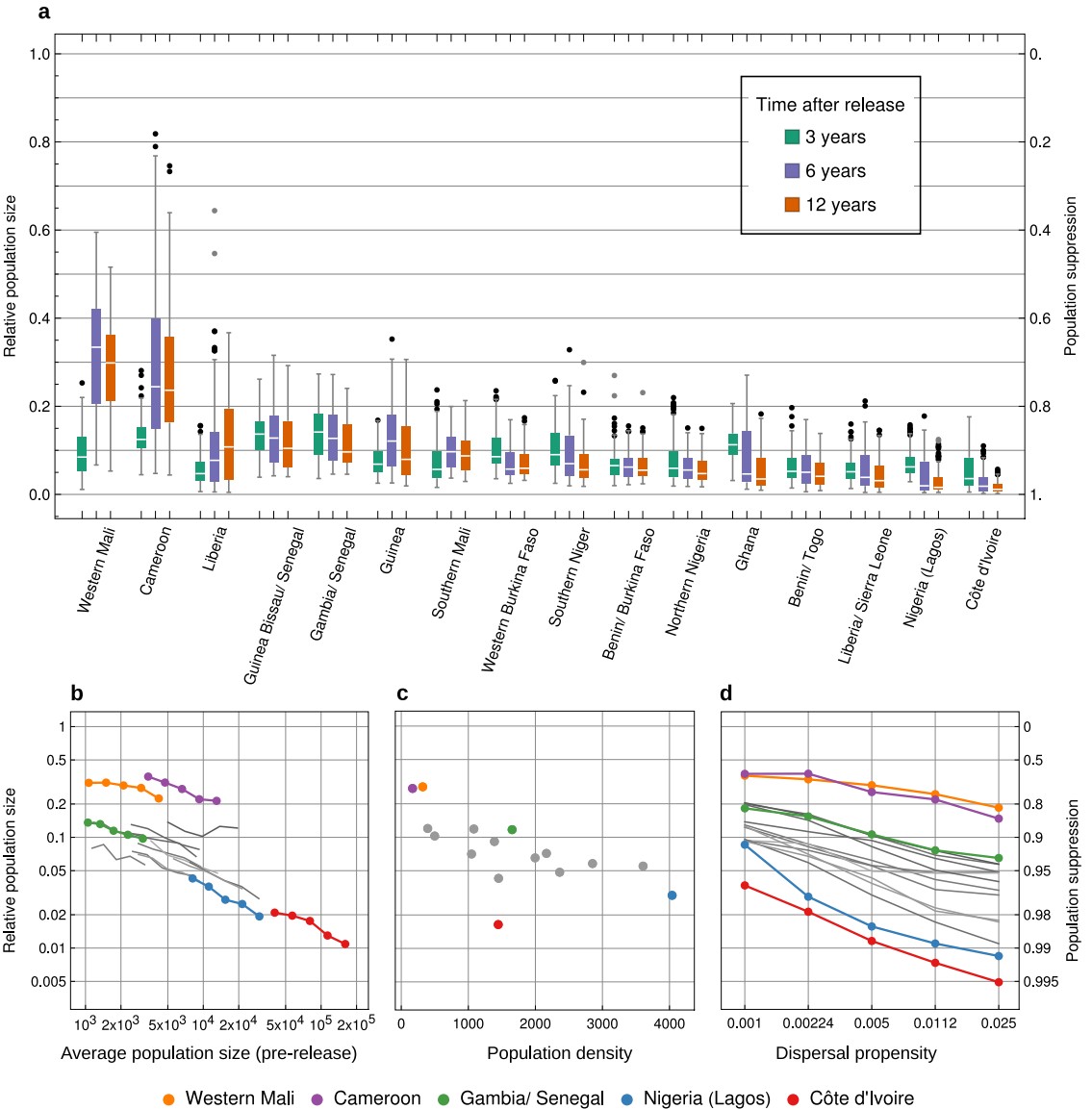

**Fig. 2 | Population suppression predicted by the mosquito metapopulation model. a** The suppression of the total number of biting females in each area in the years following gene drive releases at time zero in all vector species groups. The boxes represent the interquartile ranges across 25 parameter sets that differed in both dispersal ($\rho_n$ & $\rho_b$) and population size parameters ($K_{1,a}$ & $K_{2,a}$) (see Methods and Supplementary Material), with 5 replicate simulations per parameter set. The white lines in the boxes show the medians, the whiskers represent 1.5 times the interquartile ranges, and the individual points are outliers. **b–d**. The sensitivity of these results to (**b**) average population size, (**c**) population density (the number of populations in the simulated area), and (**d**) the mosquito dispersal propensity. In each plot, the points show average 12 year suppression across simulations sharing the same focal parameter, and we highlight the results of five simulation areas (coloured; the grey lines/points represent the remaining eleven simulation areas). All simulations followed the default release strategy, described in the text, of 1000 males released in 50 localities in all three species groups.

transmission[30], and cryptic species harbouring *Plasmodium* infections have been discovered[31] and may also contribute to transmission. Also, *An. stephensi*, an urban malaria vector, has invaded into east Africa in the last decade and has recently been found in parts of west Africa[32].

To assess how gene drive strategies can act alongside other new interventions to improve malaria control, we investigate the performance of gene drive releases alone and in combination with two other new interventions that are currently being implemented widely in Africa. Firstly, we model a rollout of the RTS,S vaccine, which has shown an efficacy of 36% against clinical malaria[33]. Secondly, we simulate switching to pyrethroid-PBO ITNs, which are next-generation insecticide-treated bednets impregnated with the synergist piperonyl butoxide (PBO) that have been shown to reduce malaria infection risk by 63% compared to pyrethroid-only ITNs[34]. We discuss how our

results can guide the design of gene drive interventions for malaria control regarding the properties of the gene drive constructs, the selection of target species and release areas, and strategies for combining gene drives with other interventions. Our analyses assume that the evolution of functional resistance to the gene drive, which would subsequently render the gene drive ineffective in malaria contol, does not occur (see the Discussion).

## Results

We model gene drive releases in sixteen areas of 1° in latitude by 1° in longitude located across Western Africa (Fig. 1a), which were chosen to span variation in a set of factors that may be important to gene drive impacts. These include the prevalence of *Plasmodium falciparum* malaria in humans, vector abundance and species composition, and

seasonality, estimated from fine-resolution geospatial layers (refs. 35–37 and see the Methods and Fig. S4). In each area, we model a mosquito metapopulation where the mosquito populations are linked to the locations of human settlements, which we estimated using fine-resolution settlement footprint data. We used a Voronoi tessellation around the settlement locations to define a locality covered by each population (e.g. Figure 1b and see Methods). Each locality has a specific human population density, rainfall profile, distribution of water bodies and vector species composition (see Methods). The estimated number of settlements/localities varied from 168 in the area located in Cameroon to 4039 in the Lagos region of Nigeria (Fig. 1a).

For each of three vector species groups (*An. funestus*, *An. arabinesis* and the combination of *An. gambiae* and *An. coluzzii*), we assume that the abundance of each population is regulated by the amount of larval habitat, which is estimated from geospatial data on weekly rainfall, the presence of rivers and lakes and the human population size. The carrying capacities of each species group were assumed to be proportional to their estimated relative abundances at each location[36]. In each area, we scaled the carrying capacities by an equal amount such that the seasonal peak number of mosquitoes in the area is consistent with that estimated by a malaria transmission dynamic model that relates mosquito abundance to the entomological inoculation rate (EIR) and the prevalence of *P. falciparum* malaria (e.g. Fig. 1b and see Methods). Among the sixteen study areas, we estimated that the largest vector populations are in the Côte d'Ivoire area ($4.2 \times 10^8$ mosquitoes across all the species groups at the seasonal peak), and the smallest are in the Cameroon area ($3.4 \times 10^6$ mosquitoes), a span of roughly two orders of magnitude.

### Impacts of gene drive releases on vector abundance

We first consider a gene drive that reduces the egg laying rate of females carrying a single copy of the gene drive by 35% compared with wild females (we later investigate sensitivity to this fitness cost). We further assume that the gene drive allele is inherited by 97.5% of the offspring of male and female heterozygotes (gene drive paired with wildtype), while the remaining offspring inherit either the wildtype allele (1.25%) or a non-functional resistant allele (R2 allele; 1.25%) due to non-homologous end-joining (NHEJ) during the homing reaction[38]. We assume females are sterile if they do not carry at least one wildtype allele.

We modelled simultaneous releases in all three vector species groups, and each simulation tracked the population dynamics for 12 years from the release date. For each area, we replicated the simulations over a broad range of mosquito movement rates and population sizes to account for uncertainty in these factors. Specifically, we varied the dispersal propensity $\rho_n$ (the probability an adult mosquito moves to a connected locality on a given day, see Methods) across five values from $\rho_n = 0.001$ to $\rho_n = 0.025$, and we varied the mosquito population size across five values from half to double the area-specific estimates (by adjusting carrying capacity parameters, see Methods), resulting in 25 parameter combinations of dispersal and population size. For each parameter combination, area, and species group, we simulated releases of 1000 gene drive heterozygous male mosquitoes in each of fifty randomly selected settlements. These releases were predicted to result in substantial reductions in biting females in all sixteen areas over 12 years (Fig. 2a). There were consistent differences between areas, however, with some areas showing average suppression at 12 years following releases of >95% (Cote d'Ivoire, Nigeria/Lagos, Liberia/Sierra Leone, Benin/Togo, Ghana, Northern Nigeria) with the remaining areas showing average suppression of >70%, with some realisations showing <50% suppression (Southern Mali and Cameroon).

### Sensitivity to population size, density, dispersal, and seasonality.

To understand these differences in suppression, we assessed the role of four factors on the simulation results described above: average

population size, population density (the number of populations per unit area), dispersal propensity, and seasonality (defined as the average number of dry weeks per year in an area; Table S2). A multivariate regression model with these predictors explained much of the variance in suppression across all simulations ($R^2 = 0.79$). There was a positive relationship between suppression and each of the first three factors, i.e. suppression is higher where mosquito populations are larger (Fig. 2b), more densely packed (Fig. 2c), and if dispersal is at the higher end of the range that we considered (Fig. 2d). Dominance analysis[39] revealed that the first three factors had similar power to explain the variance in our suppression results; population size contributed 30% of the overall $R^2$ (standardised Dominance 0.30), dispersal contributed 32%, and population density contributed 29%.

### The role of recolonisation rate.

The commonality of the above three factors—population size, population density, and dispersal propensity – is that they all affect the numbers of mosquitoes moving between populations. We contend that suppression increases with population-level mobility because higher mobility increases gene drive spread into wildtype populations, reducing the likelihood that wildtype mosquitoes recolonise habitats where local extinctions have occurred. If this explanation is correct, we can expect reduced rates of populations cycling between extinction and recolonisation as mobility increases.

To test our assertion, we computed the rate at which populations cycle through states of extinction and recolonisation for each simulation. Specifically, we defined the local suppression $s_{i,d}$ at locality $i$ on day $d$ as $s_{i,d} = 1 - \frac{n_{i,d}}{n^*_{i,d}}$ where $n_{i,d}$ is the number of biting females in that population and $n^*_{i,d}$ the equivalent number from a simulation of the same scenario except without gene drive releases. We defined a recolonisation event in locality $i$ as a transition from extinction ($s_{i,d_1} = 1$) to a state of recolonisation ($s_{i,d_2} < 0.5$) in the time-series ($s_{i,1}, s_{i,2}, \ldots$), and defined the population cycling rate to be the average number of recolonisation events per year among the populations where the gene drive was released. Each of the three explanatory variables - population size, population density, and dispersal propensity - negatively associated with population cycling rate (Fig. S1a–c). Moreover, we found a strong negative association between population cycling rate and area-wide suppression (Fig. S1d). These results support our assertion that the three explanatory variables affect suppression via their influence on population-level mobility.

### Variation in spatial dynamics.

The effects of population-level mobility on extinction and recolonisation cycling, and thus suppression, resulted in markedly different spatial dynamics in the different study areas. In areas with low mobility, the wildtype allele frequently escapes the influence of the gene drive allele to recolonise habitat where the mosquito had previously become extinct. For example, in the Western Mali area where the populations are small and widely spaced, a typical simulation reveals irregular waves of extinction followed by waves of wildtype recolonisation such that the landscape is an ever-shifting mosaic of neighbourhoods in different states (Fig. S2; a simulation of gene drive dynamics in *An. arabiensis* in Western Mali is viewable here) [https://github.com/AceRNorth/Animations/blob/main/anim201.gif]. By contrast, in regions with larger and more numerous populations, vacated habitat tends to be rapidly recolonised by neighbouring populations that contain the gene drive such that a degree of population suppression is maintained (e.g. *An. arabiensis* in the Côte d'Ivoire area; Fig. S2; animation viewable here) [https://github.com/AceRNorth/Animations/blob/main/anim211.gif]. The dynamics in most of the study areas fell between these two types. A simulation in the Southern Mali area, for example, shows that local extinctions are typically recolonised by neighbouring populations containing the gene drive, yet there are also occasions where populations become free of the gene drive and the wildtype allele temporarily spreads

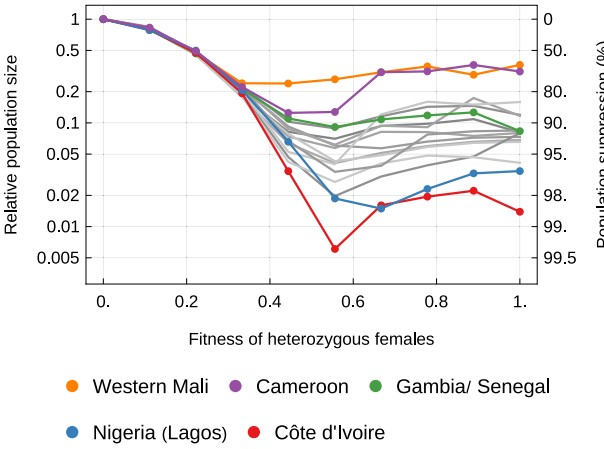

**Fig. 3 | The sensitivity of predicted population suppression to the fitness of heterozygous females.** The points represent the total suppression of all three species groups 12 years after releases in each species group, averaged across 5 replicate simulations. All other model parameters are set to their default values (Table S3).

locally (Fig. S2; animation viewable here; *An arabiensis*) [https://github.com/AceRNorth/Animations/blob/main/anim203.gif].

**Sensitivity to heterozygote gene drive fitness.** Previous studies of female fertility gene drives have highlighted the importance of the effect of the gene drive on fertility in heterozygous females[13,18], which we now consider (Fig. 3). Predicted suppression is zero in the extreme case of heterozygous sterility, because such a drive allele will not increase from rare even in a local well-mixed population[13,18]. As the fitness of heterozygous females increases from zero, the predicted 12 year suppression gradually increases in all study areas, up to a fitness of about 0.4 (meaning that one-copy females produce 60% fewer eggs than wildtype females). Indeed, up to this fitness level, the predicted suppression in all sixteen areas is near identical. As fitness increases further, the study areas diverge in predicted suppression as discussed above. Across all areas, however, suppression tends not to increase as heterozygous fitness levels increase beyond 0.5.

To understand these fitness effects we must consider how the gene drive allele affects individual populations. If the fitness of heterozygous females is low (0–0.4), the drive allele will rarely become locally fixed. Instead, it will rise to an equilibrium frequency with both the wildtype doublesex gene and R2 alleles being present. This equilibrium results in population suppression rather than extinction, to a degree that depends on the drive allele fitness[18]. Such a gene drive will thus tend to spread radially from its release rather than induce extinction and recolonisation dynamics, which is why the predicted results are similar in all the areas we consider. At fitness levels greater than ~0.4, local suppression more frequently causes local extinction, thereby causing a switch to extinction and recolonisation dynamics, whose precise form will depend on the area.

**Impacts of gene drive releases on the malaria burden**
For each of the sixteen areas, we use our malaria transmission dynamic model to compare the impacts of gene drive releases in each of the three vector species groups on the prevalence of *P. falciparum* malaria in the human population. In each area, we parameterise the model to approximate historical trends in malaria prevalence in humans over the period 2000–2018 accounting for changes in the coverage of malaria control interventions throughout this time, including vector control interventions such as ITNs and IRS, and drug treatments and SMC. This model assumes that the human population is well-mixed throughout each modelled area (see Methods). We first consider the

impacts of releases in a single vector species group only (Fig. 4; blue, green and red lines and markers show releases in *An. funestus*, *An. arabiensis*, and both *An. gambiae* and *An. coluzzii* respectively).

In the majority of areas, the greatest reductions in prevalence are found when releases occur in the *An. gambiae/An. coluzzii* group (Fig. 4; Senegal/Guinea Bissau, Ghana, Nigeria (Lagos), Cameroon, Sierra Leone/Liberia, Guinea, Togo/Benin, Liberia, Côte d'Ivoire). These areas have a high relative abundance of *An. gambiae* and *An. coluzzii* (Fig. 4). Moreover, these two vector species have higher rates of blood feeding on humans compared to *An. arabiensis*, which takes more blood meals from non-human hosts (see the parameter $Q_O$ in Table S4). Targeting these species is thus more effective in reducing transmission compared to targeting *An. arabiensis*. When releases occur in *An. gambiae/coluzzii*, the five areas with the greatest relative reductions in prevalence were 68% (95% central quantile: 37–86%) in Senegal/Guinea Bissau, 65% (CI: 51–82%) in Nigeria (Lagos), 64% (CI: 44–91%) in Ghana, 40% (CI: 29–51%) in Sierra Leone/Liberia, and 38% (CI: 28–49%) in Guinea. Larger relative reductions in prevalence occurred in areas where the malaria prevalence prior to the releases was relatively low, such as in Senegal/Guinea Bissau, Nigeria (Lagos) and Ghana because reductions in transmission have larger impacts on prevalence at lower initial prevalence levels (as discussed further below). Releases in *An. arabiensis* produced strong reductions in prevalence in Niger/Nigeria and northern Nigeria (Fig. 4a), which are northern areas where *An. arabiensis* is the predominant vector. In these two areas, pre-release malaria prevalence was low, and releasing in *An. arabiensis* was able to halt malaria transmission. Releasing in *An. funestus* was the best strategy in Western Mali, Southern Mali, Western Burkina Faso and Benin/Burkina Faso (Fig. 4).

We next compare releases in a single vector species group with releases in all three vector species groups simultaneously. The latter strategy results in much greater reductions in prevalence that exceed the additive effects of releasing in each of the three vector species groups individually. After >5 years from the release date, prevalence remains below 5% in all areas, except for Liberia and Cameroon, where prevalence remains at 5–10% (Fig. 4; dark green triangles). Strong reductions in prevalence occur when releases are made in all vector species because the relationship between malaria transmission and prevalence is non-linear, with prevalence falling away rapidly once malaria transmission drops to low levels (Fig. S3).

**Impacts of gene drive releases in different species combinations.** We now compare a range of release strategies that target different combinations of vector species to investigate the reductions in the disease burden that could be achieved, as measured by the numbers of clinical malaria cases. Here, we also consider alternative scenarios where an additional non-target vector species is present, to represent the likely possibility that other vector species may contribute to transmission. We model the impacts of releases targeting *An. gambiae* and *An. coluzzii* only (Fig. 5, bars labelled GC), *An. gambiae, An. coluzzii* and *An. arabiensis* (Fig. 5, bars labelled GCA), and all three vector species groups (Fig. 5, bars labelled GCAF). In the case where releases occur in all three vector species groups, we consider additional alternative scenarios in which a non-target vector species is present in the area with a relative abundance of 10% (GCAF*10) and 20% (GCAF*20) of the combined abundance of the three vector species groups targeted by gene drive releases. We assume that this non-target species is identical to *An. arabiensis* in terms of all demographic and behavioural parameters (see Methods). We measure impacts by the cumulative number of clinical malaria cases averted in children aged 0–5 years over the 12 year period following releases. We also calculate the cumulative number of cases occurring over the period and show the proportional as well as absolute reductions in cumulative cases in each area achieved by each release strategy (Fig. 5).

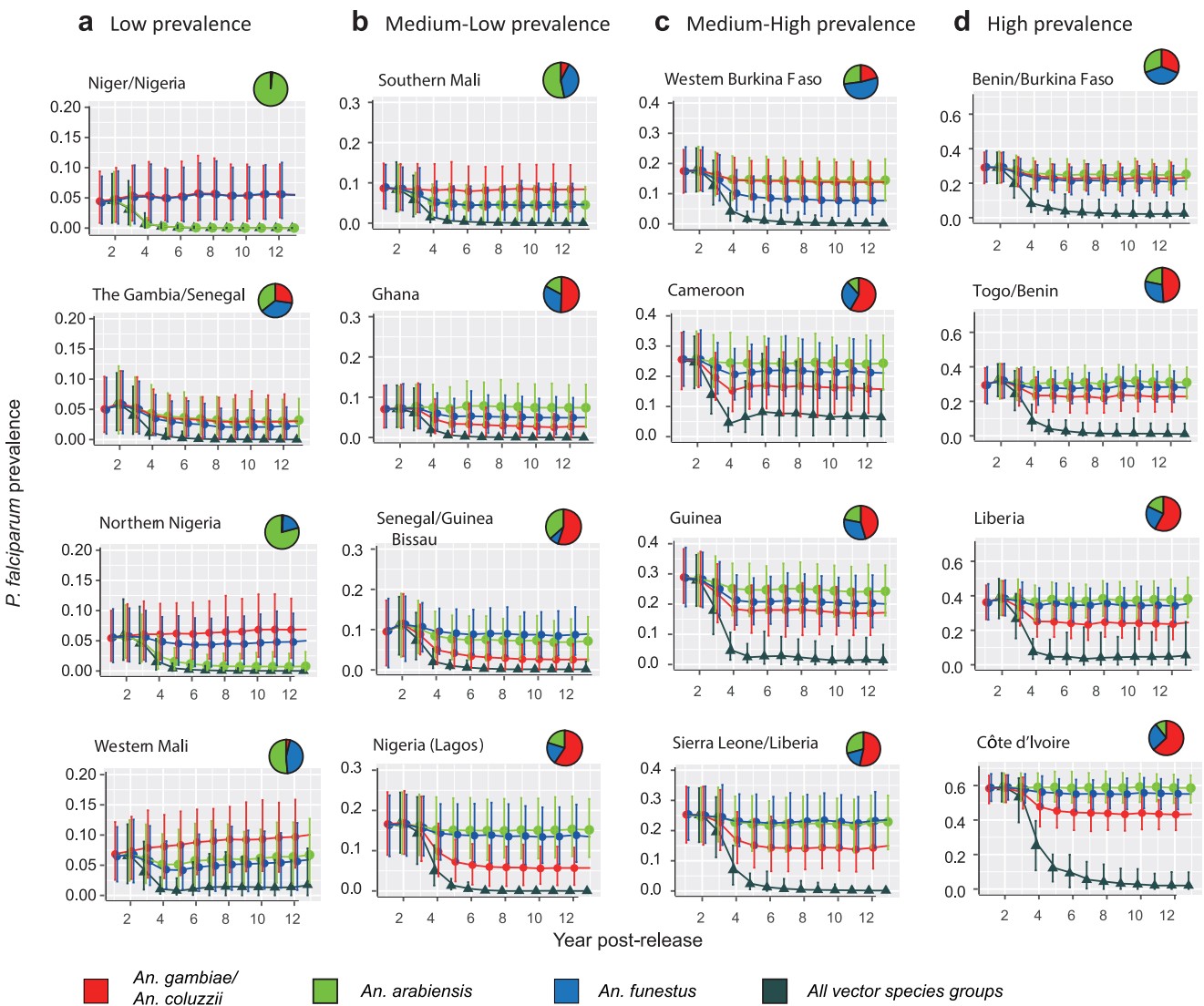

**Fig. 4 | Impacts of gene drive releases in different vector species groups on the prevalence of *P. falciparum* malaria.** Markers show the average annual prevalence of *P. falciparum* malaria in each year following gene drive releases, in either *An. arabiensis* (green), *An. funestus* (blue), both *An. gambiae* and *An. coluzzii* (red), or in all four vector species (dark green). Markers and error bars show the means and the 95% central quantiles, respectively, from 125 simulations. Pie charts show the proportion of each of the three vector species groups in each area. Columns (**a–d**) divide the sixteen areas into the quartiles of the average annual prevalence in the year prior to gene drive releases, where (**a–d**) show the first, second, third and fourth quartiles, respectively.

Gene drive releases in the *An. gambiae/An. coluzzii* species group (GC) reduced the average number of cumulative clinical cases across the sixteen areas by as much as 70% (19–100%) in Senegal/Guinea Bissau. In many areas, releasing gene drives in *An. arabiensis, An. gambiae* and *An. coluzzii* (GCA), had a modest additional benefit compared to releasing in *An. gambiae* and *An. coluzzii* only (GC) (Fig. 5a; Côte d'Ivoire, Liberia, Togo/Benin, Guinea, Cameroon, Western Burkina Faso, Nigeria (Lagos), Ghana). In these eight areas, the greatest additional benefit of releasing in *An. arabiensis* occurs in the Western Burkina Faso area where the average reduction in clinical cases increases to 37% (14–66%) from 19% (0.05–41%). In areas with lower pre-release transmission levels (Fig. 5a, top panel), there is often a much greater additional benefit to releasing in *An. arabiensis*. For example, in the Niger/Nigeria area, gene drive releases in all *An. gambiae* complex species (GCA) averted 37,728 (5630, 75,552) clinical cases per 100,000 children over the 12-year period, which is a 97% reduction in average cases compared to a scenario where no releases occurred. This release strategy also achieved large reductions in cases in northern Nigeria, Western and Southern Mali, Senegal/Guinea Bissau, and The Gambia/Senegal (Fig. 5a). This is due to the predominance of *An.*

*arabiensis* in these areas and their relatively low pre-release malaria prevalence (Fig. S3).

In most areas, releasing in all three vector species groups (GCAF) gives much greater benefit compared to releasing only in species from the *An. gambiae* complex (GCA), even when a non-target vector species is present (Fig. 5a). This strategy achieved large reductions in absolute numbers of clinical cases in the highest transmission settings, such as Côte d'Ivoire, Liberia, Togo/Benin and Benin/Burkina Faso (Fig. 5a, bottom panel). For example, in Côte d'Ivoire, 421,153 (344,678, 481,496) clinical cases per 100,000 children were averted over the 12-year period by releasing in all three vector species groups where a non-target vector species had a relative abundance of 20% (GCAF*20). This represents a 69% reduction in average cumulative cases compared to a scenario where no releases occurred. The corresponding reduction in average cumulative cases in Côte d'Ivoire achieved by this strategy when no non-target vector species were present (GCAF) was 88% (Fig. 5b; GCAF).

**Impacts of gene drives combined with new interventions.** New interventions applied concurrently with gene drive releases have the

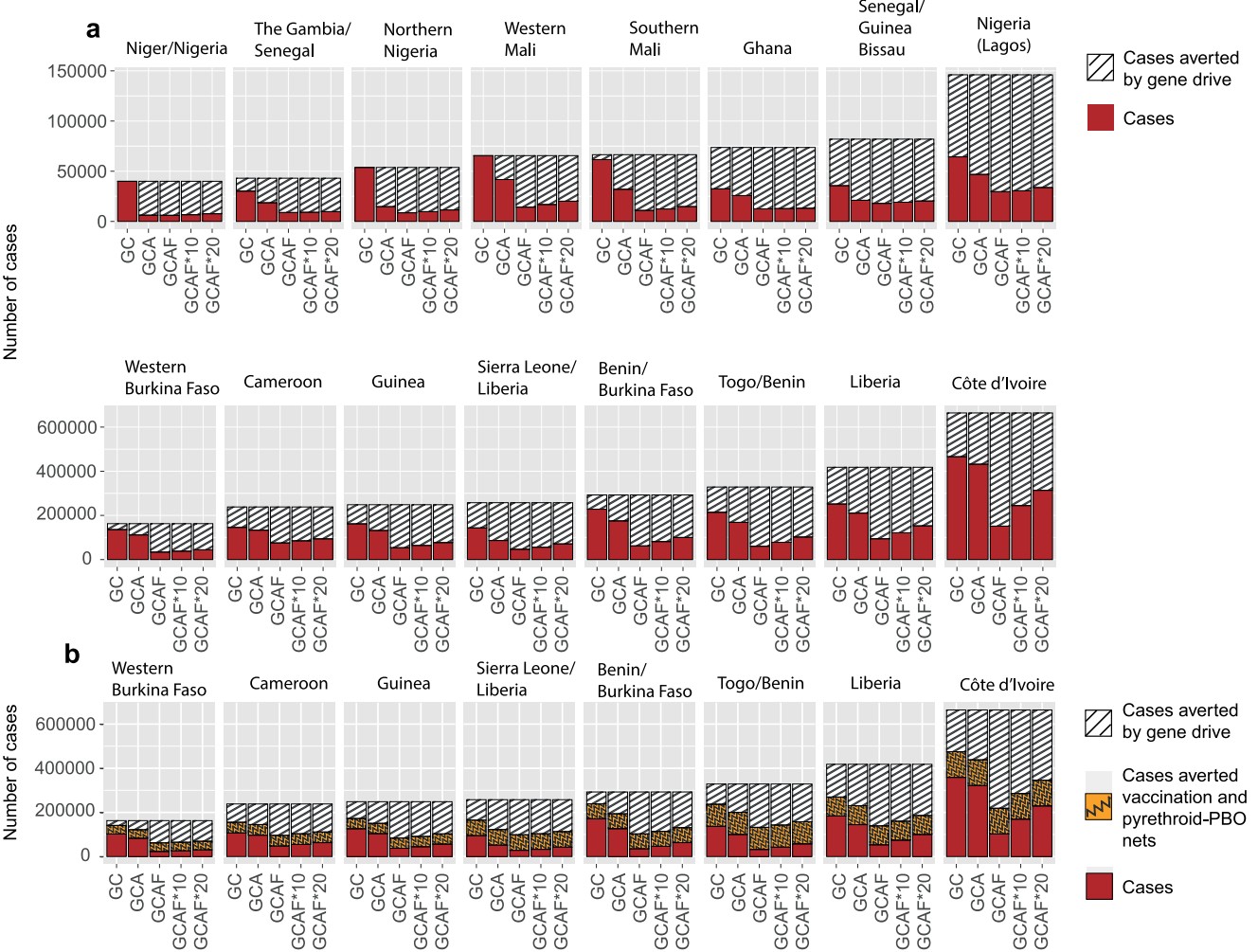

**Fig. 5 | Impacts of gene drive releases on malaria cases when implemented with and without vaccination and new types of ITNs.** Releases in three different species combinations are modelled, denoted GC, GCA and GCAF, where for the third strategy the asterisks denote the presence of other non-target vector species at different relative abundances (see the main text). **a** Red bars show the cumulative number of cases occurring in children aged 0–5 years over the 12 year period following the implementation of gene drive interventions in each area. Hashed bars show the cumulative number of cases averted by gene drive releases compared to the counterfactual scenario where no gene drive release occurred. **b** For the eight areas with the highest pre-release malaria prevalence, red bars show the cumulative number of cases occurring when gene drive releases are combined with RTS,S vaccination and switching to pyrethroid-PBO ITNs. Orange patterned bars show the cumulative number of cases averted by applying both RTS,S vaccination and pyrethroid-PBO ITNs, and hashed bars show the additional number of cases averted when gene drive releases are implemented in combination with RTS,S vaccination and pyrethroid-PBO ITNs. Data are shown per 100,000 children.

potential to reduce the malaria burden to lower levels, which will be especially critical when gene drives are not able to target all vector species. We estimate the reductions in clinical malaria cases that could be achieved by combining gene drives with two other more recent interventions, RTS,S vaccines and switching from pyrethroid-only to pyrethroid-PBO ITNs, where we assume that switching between ITN types does not change the ITN coverage across the human population in each area (see Methods). We show the impacts of combining gene drive releases with these other interventions for the top 50% of areas with the highest pre-release malaria prevalence (Fig. 5b), because the impacts of combining interventions are greatest in high transmission settings.

In most of the high transmission areas, when the vaccination and pyrethroid-PBO net interventions are in place gene drive releases produced substantial added benefit in averting clinical cases (Fig. 5b). Gene drive releases in *An. gambiae* and *An. coluzzii* (GC) increase the average number of cases averted by at least 60% (in western Burkina Faso) to as much as 170% (in Liberia) across the eight areas, relative to when vaccines and pyrethroid-PBO net interventions are applied

without gene drive releases (Fig. 5b, comparing orange patterned bars versus white hashed bars). Additional releases in *An. arabiensis* (GCA) increase the average number of cases averted by at least 116% (in western Burkina Faso) to as much as 226% (in Guinea) across areas. Releasing in all three vector species groups (GCAF) increases the average number of cases averted by at least 210% (in Togo/Benin) to as much as 397% (in Côte d'Ivoire), or by at least 180% (in Togo/Benin) to as much as 332% (in Guinea) when a non-target vector species has a relative abundance of 20% (GCAF*20). The number of cases averted by combining interventions is, however, less than the sum of the cases averted by each intervention in isolation. This is because an intervention averts more cases (in absolute terms) when case numbers are higher, therefore additional interventions lower the number of cases that are averted by each intervention.

## Discussion

Malaria is one of the most harmful infectious diseases afflicting humans and continues to cause severe mortality, especially in Sub-Saharan Africa. Progress has been made over recent decades in

reducing this burden, and new control measures have recently been introduced or are in development. Population modelling can be important in optimally deploying existing and future interventions, and is required as part of the regulatory process to assess the safety and efficacy of novel techniques such as gene drive[12]. However, there are numerous challenges to modelling a complex system involving humans, a pathogen and multiple vectors across broad geographic areas with many varying biophysical and socioeconomic factors.

We develop here a novel approach to modelling gene drive interventions for malaria control that includes entomological and epidemiological processes, and incorporates spatially-specific parameter inputs. It differs from previous models in being able to track entomological interventions through to clinical outcomes, and in its ability to study the combined effect of control measures aimed at insects and humans, as well as how local conditions affect the success of disease suppression. We use the approach to explore the potential of releasing gene drives to suppress vector populations and hence malaria prevalence in west Africa. We confirm existing results about the importance of the molecular construct's properties (in particular, fitness in the heterozygote) and of the spatial structure of local vector populations, and show that these conclusions apply in different spatial settings. We first explore the relative advantages of introducing gene drives into the most important of all known local vector groups, and then look at the interaction of gene drive with two other major current interventions.

Our results show that, in almost all the areas we studied, while gene drive releases targeting the *An. gambiae* species complex (*An. gambiae, An. coluzzii*, and *An. arabiensis*) alone can have a significant impact, targeting *An. funestus* as well gives much greater malaria control. This resulted in malaria prevalence dropping to zero in several of the areas modelled, although we note that existing approaches to gene drive modelling, including that developed here, do not account for importation of infections from outside the area (see below). Mosquitoes in the *An. gambiae* complex are the most important African malaria vectors and so have been the target of most gene-drive research. Our findings support the importance of recent research into developing *An. funestus* gene drives[40].

Though a combination of interventions provides the strongest disease suppression (see below), there will be circumstances where it is difficult to achieve a high uptake of personal protection by public health interventions[39,41]. For example, such interventions are clearly difficult in places suffering civil strife or war where gene drive relying either on natural spread or releases using UAVs (drones) may still be possible. In such settings where malaria control using existing interventions is particularly challenging, our results indicate that gene drive releases that target the multiple vector species responsible for transmission will be important to achieving strong epidemiological impacts.

While there are potential advantages of releasing a gene drive in multiple vector species, the same impacts on malaria may occur after single-species releases, though more slowly. Malaria transmission by vector species that are not gene drive targets may eventually be inhibited by interspecific spread of gene drives. A gene drive released in a single vector species is likely to spread into closely related species through hybridisation. High rates of hybridisation between *An. gambiae* and *An. coluzzii* have been observed[42], and adaptive introgression between the two species has driven rapid spread of genetic mutations conferring insecticide resistance under selective pressure resulting from the rollout of ITNs[43]. Gene drives impose strong selective pressure and could thus also spread rapidly between these sibling species. Rates of hybridisation between *An. gambiae* and *An. arabiensis* are lower but still significant, with potential for adaptive introgression for traits under selection[44].

Of the existing interventions targeting malaria vectors, ITNs have historically been responsible for the largest reductions in the malaria burden in Sub-Saharan Africa[45]. Pyrethroid-PBO nets are replacing pyrethroid-only ITNs in several African countries[46]. After many years of research there has been significant progress in malaria vaccinology, particularly the RTS,S and R21 vaccines[47], which are now being widely deployed. Our results suggest that gene drive releases can produce substantial additional malaria reductions in areas where these new interventions are in place. In some areas, greater than 100% more clinical cases were averted when control was supplemented by releases in *An. gambiae* and *An. coluzzii* only, and over 300% when gene drive was introduced into all major vector species. Thus even in areas with good public health infrastructure and widespread adoption of other interventions there may be an advantage to also deploying gene drive. We stress, of course, that gene drive technology is still at an early stage of development and further research is required to confirm it performs as assumed here and that it meets regulatory safety requirements.

Gene drive releases have high predicted epidemiological impact through suppression of the target vector species. Though we identified some variation in suppression that is dependent on local environmental characteristics, for most parameter combinations we found significant suppression in all landscape types. The highest suppression occurred in areas where mosquito populations are both locally large and spatially numerous, such as in Côte d'Ivoire and the Lagos region of Nigeria, where the gene drive tends to spread with its wildtype counterpart such that the suppression of individual populations is largely maintained. Lower impacts were predicted in landscapes with small and widely spaced populations, such as in Western Mali and Cameroon, where the wildtype allele often escapes the influence of the gene drive to recolonise habitat. These spatial dynamics have previously been described as chasing dynamics[20]. Other modelling studies have noted the importance of landscape spatial structure, similarly finding that suppression is higher where populations are more closely packed[15,16,18], though to our knowledge the significance of local population size has not been previously recognised.

We have also shown that, for gene drives targeting the *doublesex* gene, population suppression impacts are robust to the fitness of heterozygous females up to a fitness cost of around 0.6, which is consistent with results of earlier modelling analyses[18]. Unlike previous studies[16,18], we did not find a large effect of seasonality. It is possible that previous studies over-estimated the role of seasonality per se by not accounting for average local population size, which tends to be smaller in regions with longer dry seasons.

There are important aspects of gene drive dynamics not considered in this analysis. For example, we have only considered resistant alleles that do not restore female fertility. A mutant allele that was both resistant and functional would have a fitness advantage in populations where the gene drive is present and would potentially spread rapidly to fixation[13], thereby terminating population suppression. Concerns over this possibility have motivated the development of gene drive constructs robust to simple mutational failure, such as multiplexed gene drive systems incorporating CRISPR with multiple guide RNAs, which greatly reduce the rate at which such resistant alleles evolve[48–50]. Theoretical models of the evolution of resistance to gene drives emphasise an important role of mosquito population size and standing genetic variation in determining the likelihood of evolution[51,52]. Consideration of resistance evolution in spatially structured populations across different ecological contexts is beyond the scope of the present study, but will be important to address in future work.

There are also uncertainties about different aspects of mosquito ecology that may affect the outputs of our model. The density-dependent processes regulating mosquito populations are poorly understood in *Anopheles* species. Assumptions about the functional form of density dependence can strongly affect predicted population suppression and hence disease outcomes[53,54]. We also need to better understand dispersal in *Anopheles* to improve our predictions about rates of spatial spread of gene drives, and our analyses do not include possible long-range dispersal[55], which could lead to gene drive

constructs jumping across large areas to encroach into new regions. We need more understanding of what mosquitoes do during the dry season in highly seasonal environments such as the Sahel—do they aestivate, persist in local refugia, or recolonise through long distance dispersal[56]? Previous modelling has indicated that aestivation may slow gene drive spread but not greatly alter the eventual suppression in an area[18]. Finally, our analysis has not considered how suppressing a particular vector species may affect the dynamics of cohabiting species, which may include vector species not targeted by a gene drive intervention, although other modelling studies conclude that this may be important[57].

Our analysis of the epidemiological impacts of gene drive releases uses a malaria transmission dynamic model that was previously fitted to data on the relationship between the entomological inoculation rate, parasite prevalence and clinical disease from multiple sites across Africa[26,58]. This modelling approach makes several simplifying assumptions, including spatial homogeneity of transmission dynamics such that an individual's risk of infection does not depend on their spatial location (or place of residence) within a modelled area. Our analysis does not consider human movement, or transfer of infection in and out of an area through human migration, and therefore cannot accurately predict malaria elimination outcomes which are strongly dependent on rates of importation of infections[59]. The models of malaria control interventions embedded in this framework[60], including sub-models describing the effects of different types of ITNs and IRS[28], drug treatments[61] and vaccines[29] have only been tested in a small number of settings. Thus, our results are best interpreted as estimates of the variability in potential gene drive impacts across different African settings, rather than precise predictions of outcomes in particular locations.

Despite the inevitable simplifications that need to be made to model malaria dynamics over broad spatial scales, we believe that our models capture the key factors influencing the epidemiological impacts of population suppression gene drive releases. Our approach is based on Africa-wide geospatial data layers and therefore could be applied to any area in Africa. Modelling will play a critical role in deciding how these gene drives should be trialled and eventually deployed, and we hope our analysis is a useful contribution to these discussions.

## Methods

### Selecting representative environments across west Africa
We consider the impacts of gene drive releases within a rectangular region of western Africa (Fig. 1). We predicted impacts within a set of localised areas corresponding to a selection of cells from a grid that divides the region into areas of 1° in latitude by 1° in longitude (-111 km x 111 km). We selected a set of sixteen areas chosen to span the range of variation in the prevalence of *P. falciparum* malaria in humans, as well as in a set of environmental characteristics. Specifically, we aimed to capture variation in mosquito vector species composition, which is an important determinant of the impact of gene drives on that target particular mosquito species. We also aimed to capture variation in mosquito abundance and how this varies seasonally, because this may influence the spread and persistence of gene drives[15,18]. Malaria vector abundance is strongly correlated with rainfall[62] and also depends on the number of humans from whom to source blood meals. We used two variables describing rainfall patterns to represent geographic variation in the amount and seasonality of rainfall: the long term average annual rainfall and the maximum number of dry weeks in a year ([37]; Table S2). To represent variation in human population size we used estimates from WorldPop[63]. Further details about how these variables were created for each area are provided in Section S4. Sources of geospatial data are provided in Table S1 and the methods used to aggregate fine resolution geospatial data to produce area-level values are described in Table S2.

We grouped all the 1° by 1° areas within our west African region into a set of four clusters differentiating the areas with respect to the above five variables describing mosquito population characteristics, namely malaria prevalence, vector species composition, mean rainfall, maximum number of dry weeks and human population size (Section S4). Clusters were identified using hierarchical cluster analysis. We set a requirement on our set of selected areas that there must be at least one area within each of these four clusters and that there must be four areas within each quartile of the mapped malaria prevalence values covering the region (Section S4). To represent national interests in the malaria control potential of gene drive releases, we also set a requirement on our selected areas that there must be at least one area in every country within our region of western Africa, excluding countries (Chad, Mauritania and Central African Republic) whose area is mainly outside the boundary of the region. The sixteen selected areas vary widely with respect to malaria prevalence and the selected mosquito population characteristics (Fig. S4).

### Modelling gene drive dynamics and their epidemiological impacts
We developed a mathematical model to investigate the epidemiological impacts of gene drive releases in each of our selected areas, using location-specific parameter values to represent the dynamics of gene drives in mosquito populations and malaria transmission in humans. Within each area, our mechanistic model represents mosquito population dynamics at the level of human settlements using a spatial metapopulation modelling framework adapted from the approach developed by North et al.[17]. The mosquito model captures the influence of the spatial structure of mosquito vector populations on the spread and persistence of gene drives following localised releases[17]. Human infection dynamics are modelled at the level of each area using an individual-based model of malaria transmission dynamics that builds on the open-source malariasimulation package in R[60] (Fig. S5 and Section S7). Details of the modelling methodology are given below and in the Supplementary Material, and the model parameter values are described in Tables S3–S9.

### Modelling mosquito and gene drive dynamics
Here we give an overview of the mosquito metapopulation model, while we refer the reader to the Section S6 for a more detailed description. The model parameters are listed in Table S3 of the Supplementary Material. We assume that mosquito populations are located at sites of human settlements, on the basis that the mosquito species being modelled are largely anthropophilic. The settlement locations were inferred from World Settlement Footprint data ([64]; Table S1), which demarks the presence or absence of buildings at 10 m resolution across the studied area.

For each species, each population is composed of juveniles, adult males, virgin females, and mated females (we do not consider egg or pupal stages). Following North et al.[18], we model three possible alleles at the gene drive locus: a wildtype allele, the drive allele, and a non-functional allele that is resistant to driving. The simulation model tracks the daily changes to the numbers of each type of individual that result from births, deaths, and dispersal between nearby populations. All three life-cycle processes (births, deaths, and dispersal) are simulated using pseudo-random draws from appropriate probability distributions.

For each species and local population, we assume that juvenile mortality increases with the number of conspecific juveniles, on the basis that larvae compete for food. The strength of competition depends on the amount of larval habitat at the given location and time of year. In each population, we assume there is a (low) baseline amount of larval habitat that is present all year round (scaled by a factor $K_{0,a}$), an amount that depends entirely on rainfall (scaled by $K_{1,a}$), and an amount that depends on water-courses whose extent is partially

dependent on rainfall (scaled by $K_{2,a}$). To manipulate population size at the level of the 1°by 1°areas, we varied the latter two factors $K_{1,a}$ and $K_{2,a}$ in tandem (we assumed $K_{1,a} = K_{2,a}$).

The larval habitat parameters were estimated using rainfall data from the ERA5 climate reanalysis[37], the proximity to water courses (rivers and lake edges[65]); the number of humans determined from the WorldPop dataset[63] and, finally, the relative abundance of the species under consideration from species distribution data[36] (Tables S1 and S2 and Section S6.2). The resulting model of larval habitat gave rise to time-varying carrying capacities for each species and location.

The mortality rate of adult male mosquitoes is assumed to be the same for all species groups and locations, while adult female mortality rates are species and area specific due to their interactions with insecticide-based interventions, namely ITNs and IRS. Exposure to insecticides varies between vector species due to species-specific proportions of blood meals taken on humans (Table S4) and rates of human biting outdoors, indoors or in bed (Table S8). Exposure also differs between areas due to variation in the coverage of ITNs and IRS across the human population.

Females mate at most once during their lives at a rate that depends on the number of adult males in the local population. After mating, a female lays a random number of eggs each day until she dies, with randomised egg genotypes that depend on the genotypes of her and her mate.

Both male and mated female adults may move between nearby populations with distance dependent dispersal propensity. We assume dispersal is possible between populations if they are within a maximum distance apart which is set to 10 km, though below this distance the rate of dispersal is greater between nearer than farther settlement pairs. Adult mosquitoes disperse among these populations at a rate $\rho_n$ (per day per mosquito). In addition, we assume that dispersal between settlements is possible if their localities share a border irrespective of the distances between the settlements, ensuring that even in quite isolated regions there is some connectivity between neighbouring settlements. Adults disperse among populations in this way at a rate $\rho_b$ (per day per mosquito), and we fix the ratio of $\rho_n : \rho_b$ when varying the overall dispersal rate. We set $\rho_b = \rho_n / 100$ such that the majority of dispersal is distance rather than border dependent.

### Simulation area and period

While the study areas are 1° by 1° degree squares, in each case we simulated a 3° by 3° degree areas centred on the study area to allow a one degree buffer from each edge (unless this overlapped with Ocean). We included these buffer areas to minimise artefactual edge effects that might arise if we instead assumed no migration in and out of the study areas. All the reported results (e.g. of mosquito population numbers and suppression) are study-area based—we discarded the simulation data from the buffer areas.

Our simulations of gene drive and malaria transmission dynamics (described below) cover a 43 year period. During the first 10 years, all vector control and human treatment interventions are assumed to be absent, and the model reaches equilibrium levels of mosquito abundance and malaria infections. Over the following 19 years, we model time-varying coverages of malaria control interventions, as described below. We model the release of gene drive mosquitoes on day 150 of the second year of the final 14 year period to simulate gene drive impact for 12 years post release. Throughout this final period we assume that the coverages of insecticidal vector control and human treatment interventions remain constant at the values of the year 2018. We refer to the period 2000–2018 as the pre-gene drive period and the following 14-year period as the post-gene drive period.

### Modelling malaria transmission dynamics

We adapt a stochastic individual-based model of malaria transmission and infection dynamics in humans that is described in detail in previous studies[26,27,58] to represent the impacts of gene drive releases on the malaria burden in humans. This model is implemented in malariasimulation[60], which we modified to incorporate the suppression effect of gene drive releases on vector populations (code is available on GitHub). We parameterise the model to represent the epidemiological characteristics of each of the sixteen geographic areas using location-specific data on human population densities, malaria prevalence in humans, vector species composition and the impacts of human treatment and vector control interventions. In each area we aim to match model-predicted values of the average annual malaria prevalence in humans to data on average annual prevalence for each year in the pre-gene drive period 2000–2018 by adjusting the estimated total abundance of the malaria vector population in the area to give the best agreement (see below). Further details of the malaria transmission modelling methodology are provided below and in the Supplementary Material, and the parameter values used in the model are detailed in Tables S4–S9.

**Human infection and immunity.** Each area $a$ has $H_a$ humans, and we assume that $H_a$ is constant over time and is estimated using data on the number of humans residing in each area $a$[63] (Tables S1 and S2). Each individual is classified by their age (in days) and their infection status. Within each area, the human population is assumed to be well-mixed, with each individual experiencing an equal exposure to any mosquito in the area including wildtype mosquitos and those carrying gene drive constructs or non-functional resistance alleles. The probability that an individual is bitten by a mosquito is assumed to increase as they age[66]. Following a bite from an infectious mosquito, a susceptible individual can develop clinical disease or otherwise remain asymptomatic. A proportion of those who develop clinical disease will develop severe illness which is associated with an increased death rate. The probability of each of these infection types depends on the individual's level of blood-stage immunity. A description of the mathematical model formulation is provided in Section S7.1 and the model parameters are detailed in Table S4.

**Drugs for treatment and prevention.** Individuals who develop clinical disease have a probability $f_{T,a}^y$ of receiving treatment, which is informed by data on the proportion of clinical cases that seek treatment in area $a$ in year $y$ within the pre-gene drive period[67,68] (Table S1). The treatment given is either sulphadoxine-pyrimethamine and amodiaquine (SP-AQ) or artemisinin combination therapy (ACT) using the drug artemether–lumefantrine (AL), where the time-varying efficacy of each drug is parameterised according to the pharmacokinetic-pharmacodynamic (PKPD) models developed by Okell et al.[61]. For each year, the probabilities of individuals that receive treatment with either AL or SP-AQ are given by data on the recorded proportions of ACT and non-ACT treatment across all treated individuals in year $k$ in area $a$, denoted $ACT_a^k$ and $SPAQ_a^k$, respectively[67,68] (Table S1). Seasonal malaria chemoprevention (SMC) is implemented in each year by assuming that a proportion of children aged between 2 and 5 years inclusive receive a dose of SP-AQ 1 month before the highest annual peak in total vector abundance. The probability that a child receives SMC is given by data on the recorded coverage of SMC in area $a$ in year $y$, $SMC_a^y$[67] (Table S1). Details of the models of drug treatments and SMC are provided in Section S7.3 and Table S5.

**Insecticide-treated bednets (ITNs).** The probability that an individual uses a long-lasting insecticidal bednet (ITN) is given by the estimated usage of ITNs in area $a$ in year $y$, $ITN_a^y$[39]. Values of $ITN_a^y$ for each area and year within the pre-gene drive period were calculated using annual predictive maps of ITN usage[41] (Tables S1 and S2). We then estimated the annual rates at which new ITNs are distributed to the individuals residing in area $a$ that were consistent with the annual usage values using the netz package in R[69], noting that this

methodology accounted for the estimated annual rates of household net loss (Section S7.4).

Models estimating the impacts of different types of ITNs on vector mortality, including standard pyrethroid-only and pyrethroid-PBO ITNs, have been developed based on data from experimental hut trials[28,70]. Our implementation of these models is described in the Section S7.4, and the associated parameter values are detailed in Table S6. Throughout the pre-gene drive period, we assume that all ITNs are standard pyrethroid-only ITNs.

**Indoor residual spraying (IRS).** The probability that an individual resides in a house in area $a$ that received indoor residual spraying (IRS) with an insecticide of class $c$ during year $y$ within the pre-gene drive period is given by the estimated coverage of IRS across individuals in the area in year $y$, $IRS_{a,c}^{y}$ [71] (Table S1). We estimate the coverage of four classes of IRS insecticide, organochlorines, pyrethroids, carbamates and organophosphates, using the annual predictive maps of IRS coverage developed by Tangena et al.[71] (Tables S1 and S2), calculating the average coverage value across all pixels in each area $a$ for each year. Models estimating how IRS efficacy is impacted by insecticide resistance have been developed based on data from experimental hut trials[28,70]. Our implementation of these models is described in Section S7.5, and the associated parameter values are detailed in Table S7.

**RTS,S vaccination.** In line with the recommendation of the World Health Organisation, we implement a four-dose vaccination strategy in each of our modelled areas[72]. This involves giving children who reach 5 months of age a primary three-dose series, followed by a booster dose at the same time in the following year. We implement a model of RTS,S vaccination that models vaccine efficacy over time by simulating antibody decay[29]. We assume a vaccination coverage of 80% of eligible children receiving the primary doses[29] and the annual boosters (see section S7.7 and Table S9).

**Estimating vector abundances.** The estimated total vector abundance in each area was adjusted to match the model-predicted area-wide annual malaria prevalences in the human population to data values over the pre-gene drive period[35] (Tables S1 and S2), accounting for fluctuations in the coverage of human treatment, SMC and vector control interventions throughout this period. This involved extracting from the mosquito metapopulation model the total number of female juvenile mosquitoes that complete development and emerge as adults in each area on each day of the simulation period. We used these values to calculate the number of females of each vector species group emerging relative to the maximum throughout the simulation period, $\Pi_{q,a,d}$ as $\Pi_{q,a,d} = e_{q,a,d}/\hat{e}_{q,a}$. Here, $e_{q,a,d}$ is the number of females of species group $q$ emerging in area $a$ on day $d$ and $\hat{e}_{q,a}$ is the maximum number of females of species group $q$ emerging on any day throughout the simulation period. These values $\Pi_{q,a,d}$ describe seasonal variation in adult emergence, and how this is impacted by vector control interventions, including insecticide-based interventions and gene drives. We use these values as input to the malaria transmission dynamic model to represent $e_{q,a,d}$ as $e_{q,a,d} = e_a \Pi_{q,a,d}$, where $e_a$ is a constant that represents the maximum adult female emergence reached in area $a$ throughout the simulation period. We note that this assumes that $\Pi_{q,a,d}$ is independent of $e_a$.

We calibrate the malaria transmission dynamic model by adjusting $e_a$ to match model predictions of the annual average prevalence of *P. falciparum* malaria in children aged between 2 and 10 years (inclusive) for each year in the pre-gene drive period to the estimated prevalence obtained from the annual geospatial layers produced by the Malaria Atlas Project[35]. We note that the earliest year in this period predates the rollout of ITNs in Sub-Saharan Africa, and so we assume that mosquito abundance was unaffected by vector control interventions

prior to this time. Values of $e_a$ for each area are estimated using a weighted likelihood function that upweights more recent observations[73]. Further details of this methodology are given in Section S7.2.

**Reporting summary**
Further information on research design is available in the Nature Portfolio Reporting Summary linked to this article.

## Data availability
All data used in these models come from open-source repositories listed in Table S1.

## Code availability
Simulation code for running the models of gene drive dynamics and malaria infection dynamics is available on GitHub: https://github.com/AceRNorth/WestAfricaModel (doi: 10.5281/zenodo.13785414)[74] https://github.com/pahanc/malariasimulation_import_mosq (doi: 10.5281/zenodo.13789477)[75]. The code is a modification of R-package malariasimulation v1.4.3, using R v4.3.0. Plots were generated in Mathematica v14.0 and R v4.3.0.

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

## Acknowledgements
The authors would like to thank John Connolly, Silke Fuchs, Bhavin Khatri and Katie Willis for providing helpful feedback on the manuscript. This work was supported by grants from the Bill & Melinda Gates Foundation and the Open Philanthropy Project. P.A.H., P.W. and A.C.G. acknowledge funding from the MRC Centre for Global Infectious Disease Analysis (reference MR/X020258/1), funded by the UK Medical Research Council (MRC). This UK funded award is carried out in the frame of the Global Health EDCTP3 Joint Undertaking. We devote special acknowledgement to the memory of Professor J.D.M. who passed away during the preparation of this manuscript.

## Author contributions
P.A.H., A.N., A.W.L., J.D.M. Designed research; P.A.H., A.N. Contributed equally to this work; P.A.H., A.N. Developed models and analysed results; P.A.H., A.N., A.W.L. Prepared data; P.A.H., A.N., A.W.L., A.C.G., H.C.J.G., A.B., J.D.M. Interpreted model results; P.A.H., A.N. Wrote the paper; P.A.H., A.N., A.W.L., P.W., A.C.G., H.C.J.G., A.B., J.D.M. provided feedback on manuscript drafts.

## Competing interests
The authors declare no competing interests.
