## [Transparent Peer Review file · Nature Communications]

The potential of gene drives in malaria vector species to control malaria in African environments

Corresponding Author: Dr Penelope Hancock

Version 0:

Reviewer comments:

Reviewer #1

(Remarks to the Author)

The manuscript under review, "The potential of gene drive releases in malaria vector species to reduce the malaria burden in different African environments" uses a spatial metapopulation model of mosquito suppression gene drive spread paired with a model of malaria transmission to estimate the reduction in malaria cases across a diverse set of real landscapes across West Africa, where such a gene drive would potentially be released. The manuscript is an important contribution to the field, addressing important gaps in the gene drive modeling literature including the use of location-specific landscape heterogeneity data, multi-species analysis (in isolation and together), integration of both population genetic and epidemiological models, and in the context of other management practices (e.g. ITNs). All of these factors have been recently highlighted in several reviews of gene drive models as important needs for the field. It is clearly written and well documented, and overall a pleasure to read. I have a couple concerns about aspects of the analysis that felt underdeveloped or at least under-described, as well as several minor details that require addressing.

Issue 1: Connectivity

The authors discuss the importance of "connectedness" or "connectivity" several times within the manuscript, but fail to provide clarity about how this term is defined or how others should interpret such a metric. Connectivity and population- or ecological network analysis has a strong foundation of quantitative descriptions across landscape ecology and the very fuzzy use of the term here is concerning. Connectivity could be defined as the number of links among specific subpopulations, the centrality of specific nodes within the network, the existence of subnetworks or isolation among different parts of the network, or a number of other metrics. This issue is most striking on L239, where the influence of several tested factors is interpreted as affecting connectedness of the subpopulations. The only variable tested that might have a direct connection to connectivity is the number of populations per unit area but that still speaks more to density than connectivity. If "connectedness facilitates the spatial spread of gene drives" as the next sentence states, what aspect of connectivity is that referencing and how could one base other analyses on these findings? L58, L431, L486, L491 also use the term without any real detail about how it is being referenced. The methods describe that populations are "connected" if they are within 10km, but that definition is different than "connectivity" which references the entire network.

The reason this is particularly troubling is that it is difficult to parse between different population configurations. Some populations and their Voronoi partitioned area are geographically large and abut many other populations, permitting dispersal within 2 steps across large distances, but this is due to their relative isolation. Some populations are geographically small and densely clustered, but may have fewer border-to-border connections. Are both of these scenarios considered well connected? I understand that dispersal works differently for large isolated pops, but the extent to which that affects the results is unclear. I would like to see more clearly defined metrics and potentially analyses that show the effect of connectivity or the effect of tested variables on connectivity, however it is defined.

Issue 2: Chasing

The authors spend considerable time describing the importance of chasing dynamics and mention this outcome several times throughout the results, but there is no quantitative analysis of the prevalence of chasing, the extent to which it influences the outcomes compared to other variables, or how landscape structure or population connectivity (see above) influence the generation of chasing. Several times throughout the manuscript, different conditions are anecdotally described as influencing chasing (e.g. L240, L270, L489, L520), but there are no analyses to provide further detail or independent

interpretation. At L57, links between chasing and connectivity, seasonality, and inbreeding are described within the introduction but not formally tested. Given the high quality of the modeling presented, the high level of landscape detail incorporated, and the importance of chasing in determining suppression success, I would like to see a more quantitative exploration of chasing, such that one could use findings from this publication to predict the likelihood of chasing events in other locations or within other model structures.

Minor comments:

L174-175: Are there any quantitative descriptions of what constitutes "higher" or "lower suppression dynamics"? L187 describes two sites as higher suppression. Are those just examples of such a result or are they the only ones that could be considered in that category?

L228: does "populations" per unit area reference something different than "average subpopulation size" within the same sentence. Generally, clear descriptions of what is considered a "subpopulation" vs a "population" are needed, and it would be helpful to state clearly whether these terms refer to mosquitos or humans by default, with the other being stated specifically when brought up (e.g. [mosquito] populations distributed across human populations).

L303 and throughout: Is there any expectation or inclusion of competitive release of other mosquito species when only one species is targeted? Do we assume that niche space is used or ignored when the target species is reduced? If not, mention in discussion with reference to limited interspecific interactions. Is this what is meant by dynamics of cohabitating species (L530), or does that reference other species/species groups not mentioned here.

L411: Need period at the end of sentence.

L460: This is the first I've seen where hybridization is considered a potential advantage, rather than a potential risk. Prospective risk analyses often mention hybridization and thus unintended spread as an issue. It would be useful to put these statements in the context of risk analysis.

Reviewer #2

(Remarks to the Author)

Hancock et al. investigated the effect of gene drive releases in three *An.* species (complex) across sixteen sites in west Africa that were selected to represent the existing variation in west Africa in terms of habitat, human population, vector abundance, and malaria prevalence. They integrated a spatial metapopulation and human/malaria transmission model and estimated reductions in vector abundance, malaria prevalence and clinical cases in children with various release strategies and other integrated malaria management/prevention tools.

Their gene drive construct is embedded within a haplosufficient female fertility gene (*dsx*) that renders the gene nonfunctional, and at least one functional copy is needed to produce eggs in female mosquitoes (unless the fitness cost is 1). Sterile females (that lack a functional copy of *dsx*) do not blood feed, hence do not transmit malaria.

They reported 77.3-99.1% reductions in vector abundance across all areas. The impacts on *P. falciparum* prevalence depended strongly on the vector species that were targeted with the gene drive, and integrated interventions further decreased the likelihood of clinical cases.

Modelling at this large spatial scale with the complexity of the problem at hand is a herculean task, and Hancock et al.'s work has important implications. The work is undoubtedly worthy of publishing, I have one major comment that needs to be addressed and other minor comments below, hopefully the authors will find them useful for clarifying the model further.

Major comment: Given the importance of the implications of the evolution of functional resistant (*r1*) genotypes, why not include in the model as the framework already exists? I strongly recommend including it. If not, the reasoning for not including it and the discussion around implications for its omission has to be much stronger.

I. 153 Voronoi tessellation around settlement locations: Are the overall human population density and their spatial distribution reflected with these 'localities' (i.e. number of localities)? What are the characteristics of each area other than the number of settlements?

Fig. 1A. could you specify the population size parameters? Why only 25 simulations per area? 25 simulations also includes variation in dispersal and population size parameters (plus potentially with different initial release sites)? It is hard to see the effects of different parameters separately, i.e. stochasticity caused by initial releases, variability caused by dispersal and not by population growth, etc.

I. 199. The release sites are 50 settlement locations that are chosen randomly within an "area" which is a 1x1 degree patch ("white squares" in animations). The initial malaria prevalence calibration, and subsequent results relating to the relative population size reductions, as well as the gene drive frequencies, are reported for these 1x1 degree 'areas'. These 'areas' are embedded in a larger square (as can be seen in the animation), but they themselves are isolated patches, is that correct? If that is the case, these areas are essentially 16 isolated 'islands' with no immigration from outside. Could you specify the boundaries of these areas?

I. 215-251 since there is no descriptive information provided for each area, other than the number of settlements and color-coded malaria prevalence, it is hard to gauge the results. Mali and Cameroon are the two areas with the lowest number of settlements, but since we don't know the other potentially important characteristics of all sites (seasonality etc), it is hard to interpret these results.

I. 253 how many simulations are done for the sensitivity analysis? Are they another set of simulations than the ones presented in Fig 2A? How are the parameter values chosen? Were they changed one-at-a-time, or simultaneously? Again, could you specify the parameters and their ranges specifically?

Fig 3. How many simulations for each parameter combination?

I. 278 Why not include the effect of fitness cost in the sensitivity analysis with the parameters above?

I. 314 Isn't this a truism and rather mis-leading? majority of areas have higher prevalence of *An. gambiae*/*An. coluzzii* group, and greatest reductions in prevalence occurred when released in those areas? i.e. highest reduction in Niger is when *An. arabiensis* is released, it is also the most abundant species in Niger...

I. 318 Following with the previous comment, *An. funestus* also has similar rates and shows the highest reduction where it is more common (W.BF)

Fig 4. Wouldn't it be more interesting to plot 'relative' prevalence so that sites are more comparable? (supports the claim in I. 326) How many simulations?

I. 336 are the release numbers same? 1000 individuals for each species in each of the 50 localities, i.e. 3000 total? could you clarify?

I. 608, the variables are mentioned in passing 'above', could you restate what these are? I am assuming they are *P. falciparum* prevalence, vector species composition, vector abundance, seasonality, and human population?

Could you present the results for these selected variables for each area in a table (could be in supp. materials) with potentially a short description of each area, e.g. high malaria prevalence, high human population, nonseasonal, etc? A table would be handy to refer to when interpreting the figures in the results section. The only data that is presented for each area are *P. falciparum* prevalence and number of settlements. Some areas do not have same sizes, that should also be presented.

I. 644-646, as I have mentioned above, the results of this study should be taken with a grain of salt when the full implications of the evolution of functional resistant genotypes are not considered.

I. 778, the only life history variable that differs across different *An.* species is the $\pi_i(q,a,d)$ is that correct?

S2. As mentioned earlier, the presenting the clusters would be handy to interpret the results.

S3.2 Depending on the number of settlement localities and the variability of localities' sizes (area) within an area, how well the data for landscape features overlap with the localities?

S3.2.2. The dispersal is only for mosquitoes but not humans, is that correct?

S3.2.4 last line in section, equation 3?

S3.2.5 last line in section, equation 3? how many 'localities' fall within a single location on the rainfall spatial lattice? How 'fine-scaled' is your extrapolation?

S3.2.6 last line in section, equation 3? Again, how 'fine-scaled' is your extrapolation?

S3.3.1 Values in Equation 1 and table S3 do not match. (10-19, or 11-20?)

S3.3.4 I am not clear about when border dispersal is used, when there is no settlement locality with L_D radius and neighbourhood dispersal is not possible? Or both are used always? For neighborhood dispersal, wouldn't it be possible that two settlement localities (say 1 and 2) share a large border than say 1 and 3, but 1 and 3 are closer?

Table S3. Could you give a range for the parameters that are recorded as 'variable'? What were the ranges used? Could you also present the 'default' values used if they are variable?

Fig. S4. Could you please do more simulations for each area and not report the results based on 1 simulation.

Version 1:

Reviewer comments:

Reviewer #1

(Remarks to the Author)

The authors have thoughtfully considered the reviewer comments and made sufficient changes or provided sufficient explanations to satisfy any previous concerns. The current manuscript is improved with regards to clarity and cohesiveness. At this point, I support acceptance for publication.

Reviewer #2

(Remarks to the Author)

The authors have addressed my concerns and clarified them in the main text when appropriate. I was happy with the changes that the authors have done in their manuscript in response to my comments and I appreciate the further clarifications in the main text, new simulations and a new plot. Their work has important implications and would be great to see in print.

REVIEWER #1:

The manuscript under review, "The potential of gene drive releases in malaria vector species to reduce the malaria burden in different African environments" uses a spatial metapopulation model of mosquito suppression gene drive spread paired with a model of malaria transmission to estimate the reduction in malaria cases across a diverse set of real landscapes across West Africa, where such a gene drive would potentially be released. The manuscript is an important contribution to the field, addressing important gaps in the gene drive modeling literature including the use of location-specific landscape heterogeneity data, multi-species analysis (in isolation and together), integration of both population genetic and epidemiological models, and in the context of other management practices (e.g. ITNs). All of these factors have been recently highlighted in several reviews of gene drive models as important needs for the field. It is clearly written and well documented, and overall a pleasure to read. I have a couple concerns about aspects of the analysis that felt underdeveloped or at least under-described, as well as several minor details that require addressing.

(i) Issue 1: Connectivity

The authors discuss the importance of "connectedness" or "connectivity" several times within the manuscript, but fail to provide clarity about how this term is defined or how others should interpret such a metric. Connectivity and population- or ecological network analysis has a strong foundation of quantitative descriptions across landscape ecology and the very fuzzy use of the term here is concerning. Connectivity could be defined as the number of links among specific subpopulations, the centrality of specific nodes within the network, the existence of subnetworks or isolation among different parts of the network, or a number of other metrics. This issue is most striking on L239, where the influence of several tested factors is interpreted as affecting connectedness of the subpopulations. The only variable tested that might have a direct connection to connectivity is the number of populations per unit area but that still speaks more to density than connectivity. If "connectedness facilitates the spatial spread of gene drives" as the next sentence states, what aspect of connectivity is that referencing and how could one base other analyses on these findings? L58, L431, L486, L491 also use the term without any real detail about how it is being referenced. The methods describe that populations are "connected" if they are within 10km, but that definition is different than "connectivity" which references the entire network.

The reason this is particularly troubling is that it is difficult to parse between different population configurations. Some populations and their Voronoi partitioned area are geographically large and abut many other populations, permitting dispersal within 2 steps across large distances, but this is due to their relative isolation. Some populations are geographically small and densely clustered, but may have fewer border-to-border connections. Are both of these scenarios considered well connected? I understand that dispersal works differently for large isolated pops, but the extent to which that affects the results is unclear. I would like to see more clearly defined metrics and potentially

analyses that show the effect of connectivity or the effect of tested variables on connectivity, however it is defined

In our submission, we chose not to use a mathematical definition of 'connectivity' (of which there are several, as the reviewer notes). However, we recognise the reviewers concern that the use of a word with different interpretations may be problematic. We have substantially re-structured and re-written the section of results describing the entomological model simulation results (pages 8-15, lines 152-278). We have removed references to 'connectivity' and 'connectedness' throughout. We now refer to "population-level mobility", which is the rate of mosquito movements among populations, as the factor mediating the relationships between suppression and population size, the density of populations per unit area, and dispersal. Our revised analysis links higher population sizes, more densely packed populations, and higher dispersal rates to higher population-level mobility (page 11 lines 193-205). We then add a new analysis of recolonisation rates (see our response to point (iii) below), showing that rates of population extinction and recolonisation are lower in areas with higher mosquito population mobility.

(ii) To the reviewer's specific question:

"Some populations and their Voronoi partitioned area are geographically large and abut many other populations, permitting dispersal within 2 steps across large distances, but this is due to their relative isolation. Some populations are geographically small and densely clustered, but may have fewer border-to-border connections. Are both of these scenarios considered well connected?"

The simple answer is the former arrangement is less "connected" (i.e. has lower population-level mobility) than the latter, since we assume most dispersal requires the localities' centre-points are within 10km of one another. In addition we assume a low rate of dispersal between localities that share a border even if their centre-points are >10km apart. The rate of this dispersal function is 100 times less than the former type of dispersal (Table S3). As stated in the Methods (page 32 line 660), the latter form of dispersal is assumed to ensure that "even in quite isolated regions there is some connectivity between neighbouring settlements". We have made it clearer in the Methods that this form of dispersal is weaker (page 32 line 659) to say that:

In addition, we assume that dispersal between settlements is possible if their localities share a border irrespective of the distances between the settlements, ensuring that even in quite isolated regions there is some connectivity between neighbouring settlements. Adults disperse among populations in this way at a rate ρ_b (per day per mosquito), and we fix the ratio of $\rho_n:\rho_b$ when varying the overall dispersal rate. We set $\rho_b=\rho_n/100$ such that the majority of dispersal is distance rather than border dependent.

(iii) Issue 2: Chasing

The authors spend considerable time describing the importance of chasing dynamics

and mention this outcome several time throughout the results, but there is no quantitative analysis of the prevalence of chasing, the extent to which it influences the outcomes compared to other variables, or how landscape structure or population connectivity (see above) influence the generation of chasing. Several times throughout the manuscript, different conditions are anecdotally described as influencing chasing (e.g. L240, L270, L489, L520), but there are no analyses to provide further detail or independent interpretation. At L57, links between chasing and connectivity, seasonality, and inbreeding are described within the introduction but not formally tested. Given the high quality of the modeling presented, the high level of landscape detail incorporated, and the importance of chasing in determining suppression success, I would like to see a more quantitative exploration of chasing, such that one could use findings from this publication to predict the likelihood of chasing events in other locations or within other model structures.

This is a helpful comment and we recognise our submission was insufficiently clear in the discussion of spatial dynamics. While “chasing dynamics” (in the loose sense) have been described in a number of theoretical papers (>5), it has proven difficult to produce a universal mathematical definition. An exception is Champer et al (ref 20) who do give a definition, but it is not clear how this definition generalises to different settings. Given the lack of a mathematical definition, we have now removed references to ‘chasing dynamics’, and instead added a new analysis of recolonisation rate – something that can be defined and measured from any given simulation. We have restructured the Results section (pages 11-13 lines 207-249) to define a ‘recolonisation rate’ and show there is a strong correlation between suppression and recolonisation rate across study areas (page 12 lines 224-228). We use this to support our explanation of variability in suppression results. We then present visualisations of spatial dynamics for selected areas (page 12-13 lines 230-249) to illustrate how spatial propagation of the gene drive is disrupted by high recolonisation rates, relative to areas that have low recolonisation rates and thus the gene drive maintains a high frequency across the landscape. We now only refer briefly to chasing dynamics in the Discussion (page 25, line 499). We say:

“These spatial dynamics have been previously described as “chasing dynamics”²⁰.”

Minor comments:

(iv) L174-175: Are there any quantitative descriptions of what constitutes “higher” or “lower suppression dynamics”? L187 describes two sites as higher suppression. Are those just examples of such a result or are they the only ones that could be considered in that category?

The terms “higher” and “lower” suppression dynamics have also been dropped in this revision. After re-structuring the suppression results, this was no longer a useful concept.

(v) L228: does “populations” per unit area reference something different than “average subpopulation size” within the same sentence. Generally, clear descriptions of what is considered a “subpopulation” vs a “population” are needed, and it would be helpful to state clearly whether these terms refer to mosquitos or humans by default, with the other being stated specifically when brought up (e.g. [mosquito] populations distributed across human populations).

We agree that we have been guilty of inconsistency here, using ‘subpopulation’ and ‘population’ interchangeably. We have now exchanged all references to ‘subpopulation’ to the simpler ‘population’.

(vi) L303 and throughout: Is there any expectation or inclusion of competitive release of other mosquito species when only one species is targeted? Do we assume that niche space is used or ignored when the target species is reduced? If not, mention in discussion with reference to limited interspecific interactions. Is this what is meant by dynamics of cohabitating species (L530), or does that reference other species/species groups not mentioned here.

We agree this may be an important consideration, though we did not consider it here. To answer the reviewer’s question, yes this is at least one of the important types of inter-species interactions that we were referring to in that sentence (in the Discussion). We have revised the sentence (page 27 lines 536-539) to make this clearer (new part in green):

Finally, our analysis has not considered how suppressing a particular vector species may affect the dynamics of cohabiting species, which may include vector species not targeted by a gene drive intervention, although other modelling studies conclude that this may be important.

(vii) L411: Need period at the end of sentence.

Fixed.

(viii) L460: This is the first I’ve seen where hybridization is considered a potential advantage, rather than a potential risk. Prospective risk analyses often mention hybridization and thus unintended spread as an issue. It would be useful to put these statements in the context of risk analysis.

Risks of gene drive spread into non-target species have been reviewed in Connolly et al. (2021) doi.org/10.1186/s12936-021-03674-6. This paper concludes that the nine species within the *An. gambiae* complex need to be considered as target organisms of the gene drive because hybridisation can occur between these sibling species. In regards to species outside the *An. gambiae* complex, the most closely related species is *An. christyi*. Connolly et al. states that “The absence of observed gene flow between species of *An.*

gambiae and An. christyi supports the lack of any significant hybridization between these species so that, for even less closely related species of Anopheles, hybridization is considered implausible." Thus we do not think that it is appropriate to discuss gene drive spread into species that are not intended gene drive targets as a risk.

REVIEWER #2:

Hancock et al. investigated the effect of gene drive releases in three An. species (complex) across sixteen sites in west Africa that were selected to represent the existing variation in west Africa in terms of habitat, human population, vector abundance, and malaria prevalence. They integrated a spatial metapopulation and human/malaria transmission model and estimated reductions in vector abundance, malaria prevalence and clinical cases in children with various release strategies and other integrated malaria management/prevention tools.

Their gene drive construct is embedded within a haplosufficient female fertility gene (dsx) that renders the gene nonfunctional, and at least one functional copy is needed to produce eggs in female mosquitoes (unless the fitness cost is 1). Sterile females (that lack a functional copy of dsx) do not blood feed, hence do not transmit malaria.

They reported 77.3-99.1% reductions in vector abundance across all areas. The impacts on P. falciparum prevalence depended strongly on the vector species that were targeted with the gene drive, and integrated interventions further decreased the likelihood of clinical cases.

Modelling at this large spatial scale with the complexity of the problem at hand is a herculean task, and Hancock et al.'s work has important implications. The work is undoubtedly worthy of publishing, I have one major comment that needs to be addressed and other minor comments below, hopefully the authors will find them useful for clarifying the model further.

(i) Major comment: Given the importance of the implications of the evolution of functional resistant (r_1) genotypes, why not include in the model as the framework already exists? I strongly recommend including it. If not, the reasoning for not including it and the discussion around implications for its omission has to be much stronger.

We completely agree with this comment and are currently working on models of functional resistance. It is, however, a complex topic as there are different ways that resistance can arise, and different ways this can be mitigated. We estimate that to incorporate resistance into our models would approximately double the length of the paper. Also, we think it might be more appropriate to explore resistance first in a purely entomological model and then add the epidemiology, which would be a different approach to that in the present paper. As we note in the manuscript (page 26, lines 513-516), the evolution of functional resistance would quickly render the gene drive completely ineffective in controlling malaria, and analyses weighting its impact against

the other setting-specific factors explored in our study would not provide insight. We thus would prefer to keep the current scope of the paper, but in the Introduction and Discussion have expanded our mentions of the importance of resistance.

Specifically, in our revised manuscript, we have included a statement in the Introduction explaining that our models do not consider the evolution of functional resistance (page 6 line 110). In the Discussion, we have expanded the paragraph on the implications of functional resistance (page 26 lines 512-523) to refer to previous modelling work emphasising the need for a detailed consideration of natural genetic variation in predicting the timing of resistance evolution. We conclude by explaining why these detailed considerations are outside the scope of our present study.

(ii) I. 153 Voronoi tessellation around settlement locations: Are the overall human population density and their spatial distribution reflected with these 'localities' (i.e. number of localities)? What are the characteristics of each area other than the number of settlements?

The short answer is yes. The "settlement" locations were inferred from fine-resolution (10m) world settlement footprint data, whereby 1km*1km squares were classified as settlements if at least 0.5% of the square is classified as 'built up' (i.e. the number of built-up sub-cells exceeds 49 of the 10,000 sub-cells within the square). (Section 6.2.1 of the Supplementary material). The settlement locations thus represent the human distribution in an approximate way. Human population density is an additional data layer that we use to scale carrying capacities in each locality (from the "Worldpop" dataset; Section 6.2.3 of the Supplementary material).

To clarify that these factors are incorporated in the model, we have added a sentence in the Results (page 7, line 123):

Each locality has a specific human population density, rainfall profile, distribution of water bodies, and vector species composition.

(iii) Fig. 1A. could you specify the population size parameters? Why only 25 simulations per area? 25 simulations also includes variation in dispersal and population size parameters (plus potentially with different initial release sites)? It is hard to see the effects of different parameters separately, i.e. stochasticity caused by initial releases, variability caused by dispersal and not by population growth, etc.

(We think the reviewer is referring to Fig. 2A)

We have now specified in the figure legend that we use parameters $K_{1,a}$ and $K_{2,a}$ to manipulate population size at the area level. In addition, we have added a clarification in the Methods (page 31 lines 632):

In each population, we assume there is a (low) baseline amount of larval habitat that is present all year round (scaled by a factor $K_{(0,a)}$), an amount that depends entirely on

rainfall (scaled by $K_{(1,a)}$), and an amount that depends on water-courses (scaled by $K_{(2,a)}$). To manipulate population size at the level of the 1° by 1° areas, we varied the latter two factors $K_{(1,a)}$ and $K_{(2,a)}$ while keeping their ratio fixed.

In each area and for each species, we simulated 5 values of dispersal and 5 values of population size (full factorial), giving 25 parameter combinations. We have added a clarification to the text (Results, page 9 lines 164) to say that (new parts in green):

Specifically, we varied the dispersal propensity ρ_n (the probability an adult mosquito moves to a connected locality on a given day, see Methods) across five values from $\rho_n=0.001$ to $\rho_n=0.025$, and we varied the mosquito population size across five values from half to double the area-specific estimates (by adjusting carrying capacity parameters, see Methods), resulting in 25 parameter combinations of dispersal and population size.

Previously we only simulated one run for each parameter combination due to the computational effort of producing already $16 \times 3 \times 25 = 1200$ simulation runs (some of which take >24hours to run). For this revision, we have replicated these simulation runs four times, so that the results are now based on 5 replicates for each parameter set, resulting in $5 \times 1200 = 6000$ simulation runs. We have updated our all results on vector population suppression and associated reductions in malaria burden accordingly. We note that this increase in the number of simulations per parameter set has resulted in minor numeric differences in our results but no qualitative difference in the stated results, their trends, or their interpretation.

(iv) I. 199. The release sites are 50 settlement locations that are chosen randomly within an "area" which is a 1x1 degree patch ("white squares" in animations). The initial malaria prevalence calibration, and subsequent results relating to the relative population size reductions, as well as the gene drive frequencies, are reported for these 1x1 degree 'areas'. These 'areas' are embedded in a larger square (as can be seen in the animation), but they themselves are isolated patches, is that correct? If that is the case, these areas are essentially 16 isolated 'islands' with no immigration from outside. Could you specify the boundaries of these areas?

The study areas were indeed 1*1 degree squares, yet the simulation areas were 3*3 degree squares centred on the study areas to allow a one degree buffer from each edge. (Obviously this overlapped with ocean in some cases so the area was smaller). We chose to use these large buffers to minimise artefactual "edge effects" that would arise if we instead assumed no migration in and out of the study areas. All the reported results (e.g. of mosquito population numbers and suppression) are study-area based – we discarded simulation data from the buffer areas.

We have added a short paragraph in the Methods to explain this (page 33 line 668) under the heading "Simulation area and period"):

While the study areas are 1° by 1° degree squares, in each case we simulated a 3° by 3° degree areas centred on the study area to allow a one degree buffer from each edge

(unless this overlapped with Ocean). We included these buffer areas to minimise artefactual “edge effects” that might arise if we instead assumed no migration in and out of the study areas. All the reported results (e.g. of mosquito population numbers and suppression) are study-area based – we discarded the simulation data from the buffer areas.

(v) l. 215-251 since there is no descriptive information provided for each area, other than the number of settlements and color-coded malaria prevalence, it is hard to gauge the results. Mali and Cameroon are the two areas with the lowest number of settlements, but since we don't know the other potentially important characteristics of all sites (seasonality etc), it is hard to interpret these results.

We agree that presenting a summary of the selected variables for each area would aid interpretation of our results. We have made a new figure (Figure S4, Supplementary Material, page 11) designed to visually compare how the sixteen areas differ with respect to the relative values of each variable. We now refer to this Figure in the Results at page 6, line 119, stating that:

“We model gene drive releases in sixteen areas of 1° in latitude by 1° in longitude located across Western Africa (Fig. 1A), which were chosen to span variation in a set of factors that may be important to gene drive impacts. These include the prevalence of Plasmodium falciparum malaria in humans, vector abundance and species composition, and seasonality, estimated from fine-resolution geospatial layers⁽³⁷⁻³⁹⁾ and see the Methods and Fig. S4.”

And in the Methods (page 30, line 597), stating that:

“The sixteen selected areas vary widely with respect to malaria prevalence and the selected mosquito population characteristics (Fig. S4).”

(vi) l. 253 how many simulations are done for the sensitivity analysis? Are they another set of simulations than the ones presented in Fig 2A? How are the parameter values chosen? Were they changed one-at-a-time, or simultaneously? Again, could you specify the parameters and their ranges specifically?

The simulations for sensitivity to population size, population density, and dispersal (now figs 2B-D), are the same as those presented in fig 2A. We ran additional simulations for sensitivity to fitness cost (Fig 3) . In current version, we ran replicate simulations for each parameter set (of which there were 1200 for Fig. 2) 5 times.

(vii) Fig 3. How many simulations for each parameter combination?

As above, we previously reported from one simulation run for each parameter combination, owing to the computational expense of running so many simulations, but now we have replicated these runs 4 more times (5 in total for each combination).

(vii) I. 278 Why not include the effect of fitness cost in the sensitivity analysis with the parameters above?

We are unsure which line the reviewer is referring to here – L278 does not seem relevant to the comment.

We chose to investigate the sensitivity of results to fitness separately because this parameter has such a large effect that varying it alongside other parameters would obscure the effects of variation in the other parameters.

(viii) I. 314 Isn't this a truism and rather mis-leading? majority of areas have higher prevalence of *An. gambiae*/*An. coluzzii* group, and greatest reductions in prevalence occurred when released in those areas? i.e. highest reduction in Niger is when *An. arabiensis* is released, it is also the most abundant species in Niger...

In our model, *An. arabiensis* is assumed to be a less efficient vector than *An. gambiae*, *An. coluzzii* and *An. funestus*, such that it takes a lower proportion of its blood meals on human hosts, relative to non-human hosts (the parameter Q_0 in Table S4). Thus, given equal proportions of the three vector species groups, we would not expect equal reductions in malaria transmission to be obtained when comparing gene drives that target different vector species groups. In our Results, we cannot simply use the data on vector species composition to predict which of the gene drives that targets a single species group will be the most effective. In our original manuscript we make this point at line 317, stating that targeting the *An. gambiae*/*An. coluzzii* and *An. funestus* species groups is more effective than targeting *An. arabiensis* because the latter species has lower rates of blood feeding on humans. We have now modified this sentence to make this point more clearly (page 15 lines 295-299).

(ix) I. 318 Following with the previous comment, *An. funestus* also has similar rates and shows the highest reduction where it is more common (W.BF)

Please see our response to the previous comment

(x) Fig 4. Wouldn't it be more interesting to plot 'relative' prevalence so that sites are more comparable? (supports the claim in I. 326) How many simulations?

The pre-release malaria prevalence is an important variable underpinning the predicted impacts of gene drive, and varies considerably across the sixteen modelled areas. Therefore, we think that showing the pre-release prevalence on the graphs is important to interpreting the results. In the text, we report relative reductions in prevalence for different areas and target species, thus our manuscript includes information on both measures.

The number of simulations, as stated above, is equal to 5 replicate simulations per parameter set, where there are 25 parameter sets across which vector dispersal and

population size are varied.

(xi) l. 336 are the release numbers same? 1000 individuals for each species in each of the 50 localities, i.e. 3000 total? could you clarify?

Yes this is correct. We have slightly re-worded the sentence on this in the Results (page 9 line 169) to:

For each parameter combination, area, and species group, we simulated releases of 1000 gene drive heterozygous male mosquitoes in each of fifty randomly selected settlements.

(xii) l. 608, the variables are mentioned in passing 'above', could you restate what these are? I am assuming they are P. falciparum prevalence, vector species composition, vector abundance, seasonality, and human population?

Yes, that's right. In our revised manuscript, we restate these variables at line 29, page 588-589.

(xiii) Could you present the results for these selected variables for each area in a table (could be in supp. materials) with potentially a short description of each area, e.g. high malaria prevalence, high human population, nonseasonal, etc? A table would be handy to refer to when interpreting the figures in the results section. The only data that is presented for each area are P. falciparum prevalence and number of settlements. Some areas do not have same sizes, that should also be presented.

Please see our response to point (v) above, explaining that we have added a new figure (Figure S4, Supplementary Material) designed to visually compare how the sixteen areas differ with respect to the relative values of each variable.

In the case of areas where part of the 1° by 1° region overlaps the sea, aggregate values of each variable including only pixels located on land. In our revised manuscript, we note this in the footnote of Table S2 in the Supplementary material (page 10; marked with a '*').

(xiv) l. 644-646, as I have mentioned above, the results of this study should be taken with a grain of salt when the full implications of the evolution of functional resistant genotypes are not considered.

We now acknowledge more clearly the importance of extending this work to functional resistance. Please see our above response to the Reviewer's point (i) on incorporating the evolution of functional resistance into the modelling analysis.

(xv) l. 778, the only life history variable that differs across different An. species is the $\pi_{(q,a,d)}$ is that correct?

This isn't quite correct, there are multiple life history parameters that vary between mosquito species. We use different adult female mortality rates for each species and area, which is linked to the area-specific coverage of IRS and ITNs. The species-specific adult female mortality rate is determined by their rate of encounter with IRS and ITNs, which depends on parameters describing the proportion of blood meals taken on humans (Q_0 ; Table S4), and rates of biting indoors and on humans in bed (Table S8), for each species.

To clarify this point, we have expanded our explanation of species-specific life history parameters in the Methods section (page 32 line 643), to state that:

The mortality rate of adult male mosquitoes is assumed to be the same for all species groups and locations, while adult female mortality rates are species and area specific due to their interactions with insecticide-based interventions, namely ITNs and IRS. Exposure to insecticides varies between vector species due to species-specific proportions of blood meals taken on humans (Table S4) and rates of human biting outdoors, indoors or in bed (Table S8). Exposure also differs between areas due to variation in the coverage of ITNs and IRS across the human population.

(xvi) S2. As mentioned earlier, the presenting the clusters would be handy to interpret the results.

As per our response to the earlier comment, we have made a new figure (Figure S4) designed to visually compare how the sixteen areas differ with respect to the relative values of each variable.

(xvii) S3.2 Depending on the number of settlement localities and the variability of localities' sizes (area) within an area, how well the data for landscape features overlap with the localities?

The data for the landscape features are available at different spatial resolutions, which are given in Table S1. The degree of overlap depends on locality areas (which is variable) and the resolution of the landscape feature in question. For example, here localities are tightly packed together (the closest they may be is 1km apart in areas with high human density), clearly there will be many sharing the same rainfall input (which is on a ~25km resolution grid) – but in some areas a locality may have a unique rainfall input.

(xiii) S3.2.2. The dispersal is only for mosquitoes but not humans, is that correct?

Yes this is correct. In the Discussion of our original manuscript (now page 27 line 547), we stated that 'Our analysis does not consider human movement, or transfer of infection in and out of an area through human migration'.

(xix) S3.2.4 last line in section, equation 3?

Correct – there were several references to equation 1 that should be equation 3 (now corrected).

(xx) S3.2.5 last line in section, equation 3? how many 'localities' fall within a single location on the rainfall spatial lattice? How 'fine-scaled' is your extrapolation?

Please see our response to point (xvii)

(xxi) S3.2.6 last line in section, equation 3? Again, how 'fine-scaled' is your extrapolation?

As above, this will vary by the size of localities.

(xxii) S3.3.1 Values in Equation 1 and table S3 do not match. (10-19, or 11-20?)

Thank you, this has now been corrected (11-20).

(xxiii) S3.3.4 I am not clear about when border dispersal is used, when there is no settlement locality with L_D radius and neighbourhood dispersal is not possible? Or both are used always? For neighborhood dispersal, wouldn't it be possible that two settlement localities (say 1 and 2) share a large border than say 1 and 3, but 1 and 3 are closer?

Both are always used, but the border dispersal is parameterised to occur at a much lower rate than neighbourhood dispersal (100 times lower), and thus only becomes important in large areas where neighbourhood dispersal is not possible (Table S3). (The border dispersal assumption is simply to prevent any areas being completely isolated). We have made it clearer in the Methods that this form of dispersal is weaker (page 32 line 658-661) to say that:

In addition, we assume that dispersal between settlements is possible if their localities share a border irrespective of the distances between the settlements, ensuring that even in quite isolated regions there is some connectivity between neighbouring settlements. Adults disperse among populations in this way at a rate ρ_b (per day per mosquito), and we fix the ratio of $\rho_n:\rho_b$ when varying the overall dispersal rate. We set $\rho_b=\rho_n/100$ such that the majority of dispersal is distance rather than border dependent.

(xxiv) Table S3. Could you give a range for the parameters that are recorded as 'variable'? What were the ranges used? Could you also present the 'default' values used if they are variable?

We have now done this

(xxv) Fig. S4. Could you please do more simulations for each area and not report the results based on 1 simulation.

As reported above, all the simulations are now replicated 5 times.

(xxvi) The code didn't compile on MacOS.

The entomological model code has only been tested on a linux OS.